# Impact of the Coronavirus Pandemic on High-Risk Infant Follow-Up (HRIF) Programs: A Survey of Academic Programs

**DOI:** 10.3390/children8100889

**Published:** 2021-10-06

**Authors:** Sanjeet Panda, Rashmi Somu, Nathalie Maitre, Garrett Levin, Ajay Pratap Singh

**Affiliations:** 1Paul L. Foster School of Medicine, Texas Tech University Health Sciences Center, El Paso 4800 Alberta Avenue, El Paso, TX 79905, USA; rash24somu@gmail.com (R.S.); Garrett.Levin@ttuhsc.edu (G.L.); ajay.singh@ttuhsc.edu (A.P.S.); 2Harbor UCLA Medical Center, Torrance, CA 90502, USA; 3Director of Early Development and Cerebral Palsy Research, Emory University School of Medicine, Atlanta, GA 30322, USA; Nathalie.linda.maitre@emory.edu

**Keywords:** High Risk Infant Follow-Up (HRIF) Clinic, telemedicine, early childhood development, neonates, preterm neonates, coronavirus, Covid-19

## Abstract

Objective: The impact of the COVID-19 pandemic on the functioning and services of academic high-risk infant follow-up (HRIF) clinics throughout North America. Study Design: Prospective 25-question questionnaire survey through REDCAP links that was sent over 10 weeks, to 105 US and 10 Canadian programs. Finally, 59 of 105 US programs and 5 of 10 Canadian responses were analyzed using SAS version 9.4. Results: In the US, 67% of programs reported closures between 1–5 months, whereas in Canada 80% of programs closed for 1–3 months. In the US 86% of programs provided telemedicine visits and only 42.5% provided multidisciplinary HRIF telemedicine visits. We enumerated innovative approaches specifically for the conduct of Telemedicine visits, the need for the standardization of various tests and services in a telemedicine setting, and to emphasize the urgent need for more government funding to improve follow-up and developmental services to this fragile group of newborns.

## 1. Introduction

Over the past two decades, neonatal High-Risk Infant Follow-up (HRIF) has become more prevalent throughout the United States, which involves monitoring, coordinating and improving the short- and long-term outcomes of at-risk neonates. While no current guidelines or consensus statements mandate the precise criteria, schedules, or assessments for HRIF, a follow-up after hospital discharge is recommended [1,2] and HRIF programs are required for accredited neonatal fellowship programs. HRIF programs most commonly care for former preterm infants but can also include term infants with varying perinatal conditions. With the increasing survival of preterm born infants, many are being discharged home on oxygen, apnea monitors, high-calorie formulas, and nasogastric and gastrostomy feeding tubes [3]. Most are likely to experience developmental delays and problems which require standardized surveillance with evidence-based assessments and multidisciplinary teams [4]. To manage these high-risk infants after discharge, HRIF provides specialty care, either through stand-alone clinics or programs that incorporate complex care, pulmonary, neurology, therapists to provide comprehensive clinical care, parent education, and early interventions [5,6].

Severe Acute Respiratory Syndrome Coronavirus 2 (SARS-CoV-2), a novel coronavirus officially named by the World Health Organization (WHO) on 11 February 2020 [7] started a global pandemic that claimed a significant number of lives and created significant healthcare access problems. To address social distancing and safety concerns, a rapid expansion of telemedicine occurred [8,9]. The challenges of telehealth were particularly difficult to overcome for HRIF programs which relied heavily on the physical assessments of developmental progression, the evaluation of needs for early intervention services, and referral to subspecialists. The effects of the ongoing pandemic and transformational changes on HRIF are still in flux and have yet to be investigated. To address this knowledge gap, we surveyed academic high-risk infant clinic programs in the United States and Canada.

We hypothesized that pandemic conditions would result in marked changes to program functioning. Secondarily, we aimed to report adaptations to practice that could offer opportunities for innovation even after the pandemic.

## 2. Methods

We used a prospective survey study design and obtained approval from the institutional review board at Texas Tech University Health Sciences Center in El Paso, Texas.

There was no central registry of the total number of high-risk infant follow-up programs in the United States. However, the 2020 ACGME Program Requirements for Graduate Medical Education in Neonatal–Perinatal Medicine was as follows: “A sufficient number of infants must be available in the NICU Follow-Up clinic to ensure an appropriate longitudinal outpatient experience for each fellow [10]”. Therefore, we relied on the list of 105 identified academic neonatal–perinatal medicine programs from the ‘Fellowship and Residency Electronic Interactive Database (FREIDA)’ maintained by the American Medical Association. A redcap link was sent to the chief email address of either the fellowship program director and/or division, resulting in over 200 emails to the 105 US programs.

As the pandemic affected much of North America, and as Canada had robust NICU Follow-up programs throughout 10 provinces, despite travel limitations we also reached out to the Canadian neonatal network leadership to distribute the survey to HRIF program director/coordinators.

The survey was conducted through RedCap links. Only one response from each institution was recorded. The online survey was designed to address the following major themes.

(1)The demographics of HRIF clinics programs;(2)The effects of the pandemic on the conduct of HRIF clinics and solutions to some common problems;(3)The effects of the pandemic on the availability of personnel in the clinic and early intervention programs;(4)Changes to reimbursements and the standardization of telemedicine visits.

The survey included 25 questions (Appendix A) and was conducted over 10 weeks between 20 December 2020 to 10 March 2021. Weekly reminders were sent to non-responders. The survey study was approved with a waiver of informed consent by the institutional review board at Texas Tech University Health Sciences Center In El Paso, Texas. Data were stored via the confidential and HIPAA-protected TTUHSC REDCAP service.

## 3. Analysis

Only descriptive data were analyzed, using the SAS Version 9.4 platform.

## 4. Results

Of 105 US programs, 59 had unique responses by the end of the survey period; of 10 Canadian province programs, 5 responded (Vancouver, British Columbia, Manitoba, Winnipeg, Hamilton, Ontario, Montreal, Quebec, Kingston, Canada). Some respondents did not answer all of the questions. The number of respondents and the frequency of completion of the survey were also reported (Figure 1).

### 4.1. Program Characteristics

The US and Canadian programs appeared to have referral NICUs with similar characteristics. The referral criteria to the HRIF programs were all NICU admissions, all with less than 36 weeks’ birth gestation, major anomalies, or genetic conditions, HIE and/or a combination of IUGR/SGA, discharge on feeding or respiratory support, neonatal abstinence syndrome, a need for ECMO, and/or critical congenital heart defects (Table 1). The following results were separated between US and Canadian programs.

### 4.2. Impact of the Pandemic on Conditions

***United States:*** Thirty percent reported not canceling HRIF clinics at all; however, 67% reported closures of between 1–5 months. Clinic frequency often decreased compared to pre-pandemic conditions, and clinics reported up to a 42.5% decrease in patient show rates. When asked about the conduct of HRIF clinic visits at the time of replying to this survey, only the vast majority (86%) were conducting either all telehealth visits or a mix of in-person and telehealth visits. When considering only those programs utilizing telehealth, 45% conducted them as multidisciplinary visits. About half of the respondents indicated developing a standardized method for conducting their HRIF telehealth visits. Most survey respondents did not know how the level of reimbursements compensated for telehealth in HRIF programs. Of those who knew, 42.5% indicated a lower (50–90%) level of reimbursement when compared to their in-person clinics. Most respondents also had a negative perception of the changes to HRIF clinic formats (Table 2).

***Canada:*** Most (80%) programs reported closures of between 1–3 months. Only one program reported a decreased clinic frequency and a decrease in clinic patient show rates. All programs reported using hybrid models for their HRIF programs and conducted multidisciplinary telemedicine visits. Four out of five standardized their HRIF clinic visits. No site responded to questions on the level of reimbursement or if payor policies affected their decision to conduct telemedicine vs. in-person visits. All Canadian programs reported a positive perception of changes to the HRIF program format due to the pandemic. Table 2.

In both Canada and the United States, most programs reported no changes to the availability of personnel in the HRIF clinic. (Appendix A).

### 4.3. Innovative Approaches to Conducting Telemedicine Visits

***United States:*** For vital measurements via telehealth, 8% of programs either asked parents to buy equipment or discharged patients with equipment (pulse oximetry and/or weighing scale). Almost 30% conducted telehealth visits without any measurements and 20% coordinated with primary care pediatricians or other clinic visits to obtain vitals (Figure 2). Few programs choosing or mandating in-person HRIF clinic visits indicated developmental testing or parental requests as reasons for doing so. Instead, most cited miscellaneous medically related reasons: TPN dependency, new complaints, and ventilator dependency (Appendix A). Most programs reported that Early Intervention therapies were provided virtually or in a hybrid model. Figure 3.

***Canada:*** Three of the five programs reported that they were coordinating with primary care pediatricians or other clinic visits to obtain vitals and most government-funded therapies were being provided in a hybrid model at the time of the survey (Figure 3).

## 5. Discussion

Significant improvements in our understanding of early childhood development, specifically in regard to preterm neonates helped in providing the current structure and services provided in HRIF clinics. At the beginning of the 21st century, there there were increasing state, regional, and national networks specifically focused on standardizing and improving the function of HRIF clinics [11]. COVID-19 also spurred the adaptation of telemedicine to continue services in the safest way possible. Our study showed that, with a majority of programs conducting in a hybrid model, only about 56% of the programs were able to conduct multidisciplinary visits and about 60% had standardized their telemedicine visits. A majority reported standardizing intake, physical exam, and ages and stages questionnaires (ASQs), while only a few programs reported standardizing DAYC and Bayley III. We noted that multidisciplinary visits and the standardization of visits were slower and lower compared to 100% of responses from Canada that had multidisciplinary telemedicine visits and 80% that had standardized telemedicine visits. This brings to the forefront the need for national guidelines to improve and standardize high-risk clinic visits in the US. Our study also shows how HRIF programs were adapted and innovated to obtain vital measurements to monitor growth parameters, with almost a third of HRIF not requiring measurements. In an attempt to further explore standardization, we asked if programs had developed guidelines or criteria that would lead to in-person clinic visits. 

Kuppala et al. [12], in their 2012 publication, described the structure and functioning of HRIF care in academic follow-up programs, and the unstable and multiple sources of funding for HRIF clinics across the nation. A subsequent publication in 2014 by Bockli et al. [13] showed that hospitals remained the majority funder of HRIF clinics, providing up to 60% of financial support. Seventy-nine percent of US responders to our study indicated decreased reimbursement rates for telemedicine HRIF clinic visits. HRIF clinics were already struggling to have more financial stability and COVID-19-related cancellations, a decrease in clinic frequency, and lower reimbursements for telemedicine visits are expected to make this problem worse and affect the sustainability of these critical services for the sickest newborns in near future.

It was no surprise to see that the majority of programs shut down HRIF clinics during the onset of the pandemic. The most significant findings included: nearly 30% of programs in the United States decreased their frequency of clinics, and close to 43% indicated a decrease in show rates even after being close to one year into the pandemic. We speculate, that this happened for the following reasons:(1)Programs had to decrease the frequency of clinics due to a lack of personnel and or funding.(2)As the majority of HRIF clinics switched to either a hybrid or entirely virtual model, the access to quality and affordable internet services could be one of the reasons for decreasing show rates.(3)The delivery of care, such as the basics of vitals measurements are still limited through telemedicine services.(4)A provider or parental discomfort with telemedicine visits, as there is a lack of standardization and validity of the impact on outcomes.(5)The unclear medicolegal and state rules with regard to telemedicine visits, and future consequences.

Regardless of the reasons, this is an alarming trend and is bound to have an unsurmountable effect on the long-term health of at-risk infants. 

Interestingly, the majority of responders from the United States had a negative perception (*55% negative*), while Canadian counterparts were majority positive (*all five positive*). However, what we do not know is whether the changes to the working of HRIF clinics had any positive or negative effect on the long-term outcomes of high-risk infants. AAP published a workshop report more than 12 years ago [2] and, with the new changes that the pandemic has brought to the forefront, there is an urgent need for newer guidelines and support for funding these essential clinics. In contrast, the nationalized healthcare system in Canada, where these services are mostly government-funded, demonstrates a more consistent approach throughout the nation [9]. Consistency and standardization are important for patient outcomes as well as research comparisons between various clinics and states within the US. Recently, Maitre et al. showed that, by the early implementation of international guidelines for the early detection of cerebral palsy, the age of diagnosis decreased from 19.5 months to 9.5 months [14], but these guidelines were based on a physical exam. Whether this can this be accomplished during mostly non-standardized telehealth visits should be explored in the future. Additionally, specific assessments for, e.g., hand function deficits which are associated with school-age motor performance [15], cannot be performed in a telemedicine setting, hence, it seems that while telehealth offers new solutions, it really cannot replace the need for an actual patient encounter to provide the best care to high-risk premature infants.

Lastly, one more important theme that emerges is that government-funded therapy programs (Early Childhood Intervention) are being conducted either in a hybrid model or entirely virtually (Figure 4). As much of provided therapies depend on actual face-to-face interaction to assess and teach parents the specific interventions, how this shift to provide therapies remotely is affecting outcomes is unknown and demands urgent attention.

There are several limitations to our study. Out of the 134 invites, 69 (51%) responses were received; nevertheless, as can be seen in Figure 1, most of the major US states and Canadian provinces that have HRIF clinics were covered. Additionally, the demographics of the programs that responded, as shown in Table 1, are like an earlier study by Kuppala et al. [12], with >80% programs having more than 40 beds and 500 NICU admissions. As our intention, was to survey the impact of the COVID-19 pandemic, a focus on larger academic HRIF clinics was expected to give the best yield of measure in terms of innovations and changes to the pandemic. Second, some programs did not provide answers to all questions, and to the new set of an extra five questions during week three; out of 69 total responses received there were a total of 40 completed questionnaires which included these added questions. To address this issue, an extra column in Table 1 and Table 2 was added, to show percentages, which gave a clearer picture of survey results, in addition to the number of responses.

Since we only approached academic neonatal programs, there is a chance that the data collected may not truly represent national trends, but as our goal was to see the impact of the pandemic on HRIF clinics, larger academic programs were specifically a group of interest in order to understand the standardization methods in developmental testing and innovative solutions to telemedicine. Additionally, there is no national or state database of private HRIF clinics in the United States. In the future, establishing a national and state HRIF database will improve the dissemination of information and the more widespread standardization of care of preterm neonates in these clinics. We did not frame questions or ask responders to review their records before answering questions; this was conducted to make it easier and increase the response rate, but it may have led to inaccuracies in responses received. Our survey did not address the impact of COVID-19 and the related changes to HRIF clinic on the neonatal outcomes, as it is too early at this point to observe measurable short- (18–24 months) or long-term (5–8 years) differences. We hope our study serves as a baseline, to encourage the further research and monitoring of at-risk neonates, as the delivery of care has significantly changed during the pandemic and post-pandemic era. Lastly, we attempted to obtain data from the Canadian neonatal network, which comprises a total of 10 provinces, by participating in sites by sharing a link to this survey with Canadian neonatal network coordinators [16,17], but we received responses from only five sites. A recent Canadian study by Albaghli et al. [15] showed that care in Canada became more consistent after the formation of the Canadian Neonatal Follow-up Network (CNFUN) in 2010, with >90% of sites now providing Bayley-III at 18 months assessment. Although the data were specific to the pre-pandemic era, it will be interesting to see more extensive reporting from CNFUN on the impact and adaptations of their clinics due to the COVID-19 pandemic.

## 6. Conclusions

This ongoing pandemic has upended the delivery of healthcare across the whole world and the care of premature high-risk infants is no exception. However, this group does represent the most vulnerable group of infants and children. Many innovative strategies developed can be utilized in the post-pandemic world to optimize HRIF clinic services and provide them in a more financially viable way. High-Risk Infants Clinic programs act like safety net programs for these special infants. We need to urgently analyze the effects of these rapidly evolving changes on the long-term health of these preterm infants. There is an urgent need to modernize our approach to the care of these infants, standardize telemedicine visits, advocate at the state/national level, and secure adequate funding for the care of all premature infants.

## Figures and Tables

**Figure 1 children-08-00889-f001:**
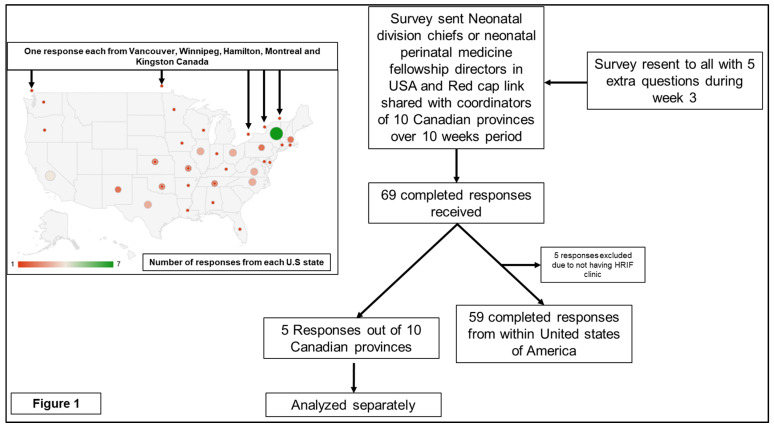
Geographical location and response breakdown.

**Figure 2 children-08-00889-f002:**
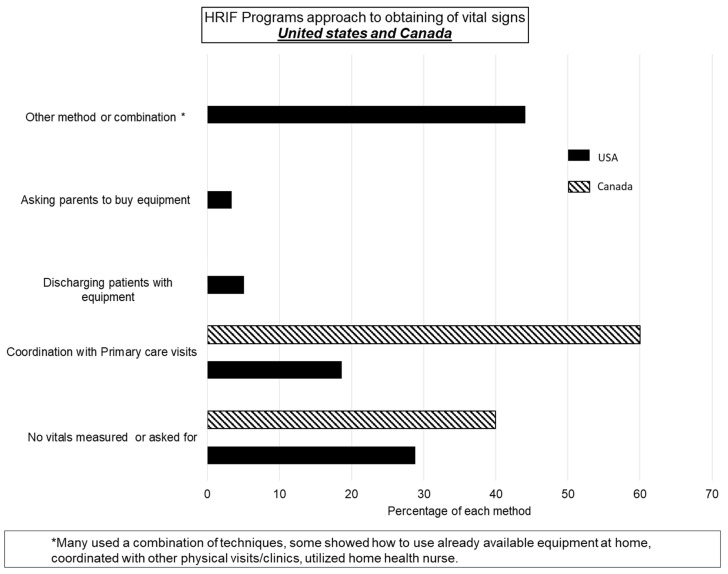
Approach to vitals measurements in HRIF clinic at the time of filling in the survey.

**Figure 3 children-08-00889-f003:**
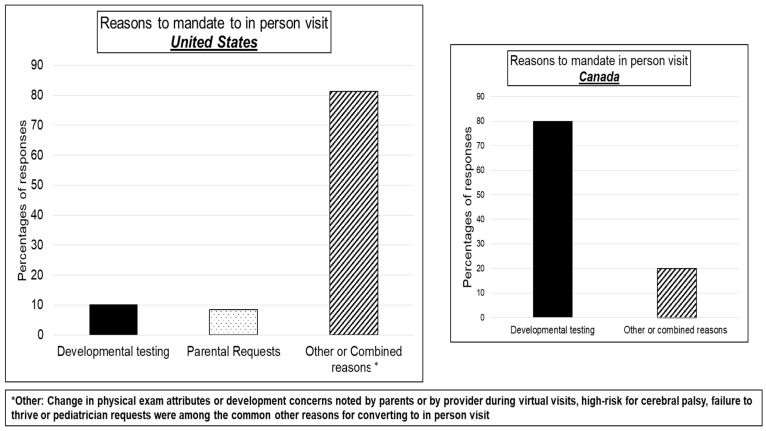
Criteria to mandate in-person HRIF clinic visits.

**Figure 4 children-08-00889-f004:**
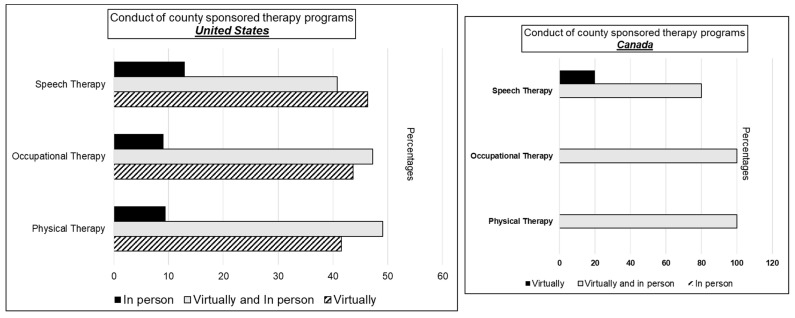
Conduct of government-funded therapy programs.

**Table 1 children-08-00889-t001:** Program characteristics.

High Risk Infant Programs Characteristics				
	United States		Canada	
Characteristic	N	Percentage	N	Percentage
	(Number of respondents)		(Number of respondents)	
Institutions with more than 40 NICU beds	59	86%	5	100
		(N = 51)		
Median numbers of Beds in NICU		60 (50–81)	5	65 (60–70)
NICU admissions per year				
<250	2	3.20%	-	-
251–500	7	11.1%	-	-
501–750	14	22.2%	1	20%
751–1000	24	38.1%	-	-
>1000	16	25.4%	4	80%
Criteria for Referral to HRIF clinic				
All discharges	1	1.60%	-	100%
All less than 36 weeks gestation	57	90%	5	100%
Genetic anomalies	36	57%	5	100%
HIE	56	89%	5	100%
Neurological disorder	43	68%	5	-
Major Malformations	40	64%	-	30%
Others *	31	49%	3	
Frequency of HRIF clinic pre-pandemic				
One Half-day per Week	14	23%	1	20%
One full day per week	14	23%	-	-
Two full days per week	10	18.5%	1	20%
More than two days per week	21	35.5%	3	75%
Average Census Per HRIF clinic day				
Less than 5 patients	8	14%	1	20%
5–10 Patients	28	47%	3	75%
10–20 Patients	19	32%	1	25%
More than 20 Patients	4	7%	-	-

* IUGR/SGA, congenital anomalies needing intervention, discharge on feeding or respiratory support, Neonatal abstinence syndrome, need for ECMO, and/or critical congenital heart defects.

**Table 2 children-08-00889-t002:** Impact of COVID-19 Pandemic on HRIF Program services.

COVID-19 Impact on HRIF Clinic Services				
Characteristic	United States		Canada	
Any Cancellation of HRIF Clinic	N	Percentage	N	Percentage
	(Number of respondents)		(Number of respondents)	
No Cancellation	19	30%	1	20%
For Less than 1 month	16	25%	3	60%
1–3 months	24	38%	1	20%
3–5 months	3	5%	-	-
>6 months	1	1.60%	-	-
Change in clinic frequency				
No Change	42	71%	4	80%
Decreased	17	29%	1	20%
Change in Patient show rates				
Increased show rates	9	15%	1	20%
Decreased show rates	25	42.5%	1	20%
No change	25	42.5%	3	60%
**Conduct of HRIF Clinic**				
All in person visits	9	14%	-	-
All Telemedicine	5	8%	-	-
Both in person and telemedicine	49	78%	5	100%
Multidisciplinary Telemedicine Visits				
Yes	18	45%	5	100%
No	15	37.5%	-	
Did not answer	7	17.5%	-	
Standardization of telemedicine visits				
Yes	21	53%	4	80%
No	13	33%	1	20%
Did not answer	6	15%	-	-
Reimbursement compared to in person visits				
<50%	4	10%	N/A	N/A
50–90%	13	32.5%		
100%	5	12.5%		
Did not know level of reimbursements	2	5%		
Did not answer	16	40%		
Did payor policies affect decision to conduct telemedicine vs in person visits?				
Yes, It affected to conducted more tele visits	5	8.50%	N/A	N/A
No, It did not affect	31	52.5		
Did not answer	23	39%		
Program’s Perception of changes to HRIF clinic format. (Sliding scale)				
Mostly negative (a score of <50)	27	55%	0	0%
Mostly positive (a score of >50)	22	45%	5	100%

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
