# Peer review of "Impact of the Coronavirus Pandemic on High-Risk Infant Follow-Up (HRIF) Programs: A Survey of Academic Programs"

_children, 2021, doi:10.3390/children8100889_

Round 1

Reviewer 1 Report

Interesting and timely paper about Impact of coronavirus pandemic on high-risk infant follow-up (HRIF) programs in the United States. About 50% of contacted hospitals sent an answer to the survey. I am curious to know if there is any significant difference between the responders and non responders in term of demographics. 

Data are well presented and the conclusions are supported by data presented.

Author Response

Dear Reviewer, 

We appreciate your time, expertise, and comments on our manuscript. In response to your comment, we elaborated in our discussion section that the demographics of the programs that responded to our study are similar to an earlier study by Kuppala et. al (12) published in 2012 with a larger respondent number and 84% response rate. Which makes our results comparable. Also, the respondents in our study represented a wide geographical spread as shown in Fig 1 of our study, which speaks to the wide reach of our survey respondents. 

Thanks

Reviewer 2 Report

Thank you for the opportunity to review this manuscript. It is an important topic, as are all specialized services that have been postponed/delayed due to the pandemic. It is good to see pediatrics investigated for this sub-set of at-risk patients and their families. The paper is a descriptive study that simply lists the results of a survey to a very segmented part of the industry. The authors do a good job listing these limitations in the manuscript, however - findings do not seem to truly help the industry improve upon current conditions. Many of the descriptive results are able to be presumed by the reader, and several assumptions are listed in the manuscript that should be further investigated to strengthen the study. Finally, the questionnaire did not report outcomes, which is an extremely important variable that readers will question, based upon the observed delivery of care methods identified. I strongly recommend a follow-up survey (if it can even be accommodated at this point since it appears one has already been conducted?) that focuses on identifying outcome trends related to such telehealth/other measures for these patients.

Author Response

Dear Reviewer, 

Thanks for your time, expertise, and comments on our manuscript. We agree with your comments. As the survey is already over, as per your recommendation we plan to do a subsequent follow-up survey in 1-2 years to monitor the changes, innovations due to pandemic and specifically look at the impact on outcomes. In our conclusion section, we have stressed following the neonatal outcomes and now to our discussion, we have added the following, 

"Our survey did not address the impact of COVID -19 and related changes to HRIF clinic on the neonatal outcomes, as it is too early at this point to see measurable short (18-24 months) or long term (5-8 years) differences. We hope our study serves as a baseline, to encourage further research and monitoring of at-risk neonates, as delivery of care has significantly changed during the pandemic and post-pandemic era."

We also have revised the discussion further in other sections to address the limitations of our study, as can be seen in the attached revised manuscript. 

Thanks again for your comments. 

Reviewer 3 Report

In this paper the authors provided data about the possible influence of SARS-COV2 on HRIF programs. This theme is actual and interesting since it points out how this pandemic condition also influenced important aspects of public health. Though interesting this paper must be reviewed with particular attention to some specific points:

1) why did they consider only academic centres? It must be explained 

2) Table 2 is somehow difficult to clearly understand. Author should decide  to consider only the complete answers (in the US part) otherwise it could be difficult to have a real picture of the situation

3) it is difficult to compare two conditions with such a discrepancy of sample size. Authors should increase the data from Canada in order to sketch also a statystical analysis which is not merely descriptive. 

4) in the discussion section the limits of the study must be better highlighted

Author Response

Dear Reviewer, 

Thanks for your time, expertise, and comments on our study. We have tried to address all your comments, in the following way, 

  1. For only academic centers, we added in discussion - Since we only approached academic neonatal programs, there is a chance that data collected may not truly represent national trends, but as our goal was to see the impact of the pandemic on HRIF clinics, larger academic programs were specifically a group of interest, to understand the standardization methods in developmental testing and innovative solutions to telemedicine. Also, there is no national or state database of private HRIF clinics in the United States. In the future establishing, a national and state HRIF database will improve the dissemination of information and more widespread standardization of care of preterm neonates in these clinics.
  2. For comments 2 and 3 -Due to variation in responses, added to ur discussion - 

    "To address this issue, an extra column in Tables 1 and 2 was added, to show percentages, which give a clearer picture of survey results, in addition to the number of responses."

     4. Revised the whole discussion section to better highlight and address the limitations of the study, as can be seen in our revised manuscript. 

Thanks

Round 2

Reviewer 2 Report

thank you for making updates based on prior suggestions.

Reviewer 3 Report

Authors adresses most of my previous comments. Therefore, according to me, it can be considered for publication in the present form